# Influence of an oral health promotion program on the evolution of dental status in New Caledonia: A focus on health inequities

Amal Skandrani[1], Helene Pichot[2], Estelle Pegon-Machat[1], Bruno Pereira[3], Stephanie Tubert-Jeannin[1] *

1 Université Clermont Auvergne, CROC, F-63000, Clermont-Ferrand, France, 2 Health and Social Agency of New Caledonia (ASS-NC), Nouméa, New Caledonia, France, 3 CHU of Clermont-Ferrand, Clinical Research and Innovation Direction (DRCI), F-63003, Clermont-Ferrand, France

* stephanie.tubert@uca.fr

**Data Availability Statement:** All relevant data are available at: https://zenodo.org/record/7148311.

**Funding:** The author(s) received no specific funding for this work.

## Abstract

New Caledonia is a *sui generis* collectivity of overseas France situated in the south Pacific Ocean. Geographical and social inequalities are superimposed on ethnic disparities with high prevalence of chronic diseases such as oral diseases. In 2012, the health agency has evaluated the children's health status. Then, an oral health promotion program was developed in 2014. Another study was conducted in 2019 in New Caledonia to appreciate the evolution of children's oral health. A sample of 488 9-years-old children was randomly selected. Dental status was clinically recorded, families and children answered questionnaires about oral health determinants. The methodology (sampling, study variables...) was similar to the one used in the 2012 study. Multivariate mixed-models were conducted to compare 2012 and 2019 dental status and to explore the determinants of caries experience in 2019. Results indicated that caries prevalence and experience decreased between 2012 and 2019, with nonetheless various trends depending on the province or type of indexes. The number of carious lesions ($d_3t + D_3T$) in 2019 was used as an outcome variable in four models. Model 1 integrated social variables; ethnicity was found to be the only significant determinant. Model 2 was related to oral health care; participation in the program & and access to oral health care was found to be significant. For oral health behaviours (model 3), tooth brushing frequency and consumption of sugary snacks were significant risk factors. In a final model with significant variables from the previous models, ethnicity, accessibility of oral health care, number of sealed molars, consumption of sugary snacks remained explanatory factors. Five years after the implementation of the oral health promotion program, positive changes in oral health have been observed. However, health equity is still an issue with varying health status depending on ethnicity, behavioural factors and accessibility to oral health care.

**Competing interests:** The authors have declared that no competing interests exist.

# Introduction

Social inequalities refer to unjust and avoidable differences in health between people or population subgroups [1]. From a social justice perspective, reducing social inequalities in health has become a major public health aim, linked to improving overall health [2]. Indeed, health promotion recognises that it is essential to act on the social determinants of health in order to provide the necessary conditions to "empower individuals to better control and improve their own health" [3].

Oral health is a marker of social inequalities in health, since early childhood. Indeed, the links between socio-economic status (income, occupation, educational level) or ethnicity and the occurrence of oral diseases (prevalence & experience) is well documented in the literature [4]. In the absence of prevention or treatment, untreated oral diseases persist into adolescence and adulthood, with a cumulative effect that accentuates the social gradient over time [5].

In 2021, the World Health Assembly approved a resolution urging WHO Member States to address determinants of oral diseases in connection with the prevention of other noncommunicable diseases. This resolution integrates a global strategy for tackling oral diseases with an action plan, that is in particular supporting health promotion interventions on oral health [6].

In order to reduce oral health inequalities, oral health promotion should be integrated in health promotion programs with a comprehensive health approach [7]. Health promotion interventions also are intended to be universal, ideally with implementation levels proportionate to the level of disadvantage (proportionate universalism) [8]. Indeed, the social gradient in the impacts of health promotion interventions vary depending on participation, acceptability or adoption of health messages related to social determinants [2]. Therefore, health inequalities have to be considered, from the development to the evaluation of health promotion interventions to ensure the ethical dimension of health promotion interventions with a search for health equity.

New Caledonia (NC) is a French territory that is following since 1998 a transition process for increased political autonomy. New Caledonia is divided into three provinces: North, South and the Loyalty Islands. The population of NC is multi-ethnic with 39% of the population declaring to belong to the Kanak community (indigenous "Oceanian" population), 27% being European and 8% Polynesian [9]. The population is very unevenly distributed between the provinces, with large socio-economic and ethnic interprovincial gaps. In the Loyalty islands, 94% of the inhabitants belong to the Kanak community, in contrast to 26% in the south Province. Furthermore, in 2019, the relative poverty rate was estimated at 18% in NC, with 9% for people declaring themselves non-Kanak, compared to 32% for Kanaks [10].

An epidemiological study assessing the oral health of New Caledonian 6, 9, 12-years-old children was conducted in 2012. The results showed that the prevalence of dental caries was high; 60% of 6- and 9-yr-old children had at least one tooth with untreated caries, the mean DMFT (Decayed, Missing and Filled permanent Teeth) was 2,09 (2,82) in 12-years-old children. In addition, important geographical, social and ethnic health disparities were identified. As an example, at age 12, oral diseases were observed in 21% of the children in the south province, 40% of children in the islands and 43% in the north province [11].

Subsequently, an oral health promotion program ("My teeth, my health"), coordinated by the NC health agency (ASS-NC) was implemented in 2014. The program is multi-sectoral, developed in connection with other health programs based on a common risk factor approach. The program targets the entire New Caledonian population with three overarching objectives: 1) (oral) health promotion in schools for better educational achievement; 2) the promotion of healthy lifestyles (oral hygiene or nutrition) such as the implementation of tooth brushing in primary schools; and 3) the reorientation of oral health services toward providing more effective and preventive interventions (dental sealants) [12,13].

In order to appreciate the evolution of oral health status, another cross-sectional epidemiological study was conducted in 2019, within a representative sample of 6, 9, 12-years-old schoolchildren. This manuscript focuses on 9-years-old children, who benefited from the program since start, with the objective of identifying the determinants and risk factors explaining the presence of untreated dental caries in 2019. Indeed, another objective was to allow the comparison of the prevalence and experience of dental caries between 2012 and 2019, in order to identify potential changes that occurred in the three NC provinces. The manuscript follows the STROBE standards for reporting observational studies.

## Population and methods

### Study population (2019)

This study was conducted among a representative sample of 9-years-old schoolchildren in New Caledonia in 2019.

- *Sample size calculation*

The sample size was estimated to have sufficient statistical power to (i) estimate the prevalence of untreated dental caries in 2019 and (ii) identify the determinants and risk factors explaining the presence of untreated dental caries in 2019. First, the sample size (N) was calculated using the formula $N = 1.96*P(1-P)/i2$, with (P) the prevalence of dental caries as assessed in 2012 (40%), (i) the margin of error fixed at 5%, and (α) the first species risk fixed at 0.05. The estimated sample size was then 184. To account for cluster effect, the sample size was multiplied by a factor of [1+ ((m-1) ICC)], where m is the mean number of children per school, and ICC is the inter-class coefficient set at 0.05 [11]. Considering an estimated participation rate of 85%, the estimated sample size was multiplied by a factor of 1.15. The number of children to be selected was 486. Otherwise, various recommendations have been proposed to determine the number of subjects required to conduct multiple regression analyses with subjects/variable ratios ranging from 15:1 to 30:1 [14]. Thus, we considered that having at least 300 children was sufficient to identify at least ten variables associated with the presence of dental caries.

- *Sampling method*

A random, stratified, cluster sampling technique was used. The study population (All 9 years old schoolchildren in NC in 2019) was stratified according to the province (Islands, North, South), the school area (Islands: Lifou, Maré, Ouvéa; North: Grand Nord, North-East, North-West; South: South- East, South-West) and the type of school (public, private). Stratification aimed to ensure representativeness in term of cultural, ethnic, geographical and social diversity. The clusters are made up of primary schools and were randomly selected with probability proportional to size. Therefore, 15 public schools and 11 private schools were selected. The total sample size was 488 in line with the required sample size.

### Ethical concerns (2019)

Ethical approval was obtained from the NC Ethics Committee (2019–06 002, 24 June 2019). Schools were approached through the local education authorities. The protocol was presented and discussed within a multisectoral committee, including associations of parents. An information letter and a consent form were sent to the families of the selected children. Only children who returned a signed parental consent form and gave verbal consent were included in the study. A declaration of conformity was also obtained from the "National Commission of Informatics and Liberty" (CNIL—MR-001- 2226401 v0).

## Study variables and data collection (2019)

Clinical examinations allowed to evaluate oral health status comprehensively but this manuscript only relates to carious status. The ICDAS classification (International Caries Detection and Assessment System) was used and the threshold for identifying dentinal lesions ($D_3T/d_3t$) was set at ICDAS grade 4 and more [15]. The presence of dental sealants on permanent molars was also recorded with the visual and tactile evaluation of the presence of sealant material (no sign vs some part vs all the occlusal fissures with sealant material) [13]. Dental examinations were performed at school. The investigators used disposable examination equipment with adequate lightening and drying devices.

Children's health determinants were assessed using an interviewer-administered "child" questionnaire, with the addition of a "parent/ adult responsible for the child " questionnaire. Questions concerned socio-demographic characteristics & living conditions, access to oral health care and prevention, oral health related behaviours. Some items (gender, date of birth, province of residence, and type of school) were directly obtained from the school register.

For identifying dental caries determinants, the dependent variable was the number of untreated carious lesions for permanent and deciduous teeth ($D_3T+ d_3t$) while other variables were considered as explanatory factors grouped in three categories (socio-demographic, oral health care access, oral health behaviours). The list of the explanatory variables is presented in S1 Table.

- *Investigators and calibration*

Six dentists performed the clinical examinations. They attended previously a two-day training and calibration seminar as in the 2012 study [11]. Inter- and intra-examiner reproducibility was checked in a sub-sample of 20 schoolchildren. Cohen's Kappa coefficients (k) showed moderate (k>0,4) to high (k>0,8) agreement for carious detection. Differences observed in caries detection were concerning mainly inter-examiner variations for initial stages (1 to 3), they did not change much caries indexes calculations. Dental assistants were also recruited to assist the dentists and/or help in administering questionnaires to children.

## Statistical analyses

- *Descriptive analysis (2019)*

A double entry was carried out and discrepancies were checked. Statistical analyses were performed using Stata software (version 16, Stat/IC, StataCorp, College Station, US) and RStudio (RStudio, PBC Version 1.4.1106). p-values < 0·05 were considered to be statistically significant. A descriptive analysis was conducted to describe the prevalence (% of children with disease) and cumulative experience of carious disease (mean caries indexes) and to present the characteristics of the study population. Results were expressed with means and standard deviations for caries indexes and percentages with 95% Confidence Intervals (CI) for other variables. For some explanatory variables, answering categories were grouped to facilitate the analyses depending on the number of answers and pertinence. The description of the study population for the explanatory variables by province and by ethnicity was analysed using chi-squared test.

- *Comparison of dental status between 2012 and 2019*

The comparison of dental status between 2012 and 2019 was performed using both initial raw data sets. Indeed, in 2012, a representative sample of 789 9-years-old children had also been randomly selected. Dental status had been clinically recorded, families and children had answered questionnaires about oral health determinants. The methodology (sampling, study variables. . .) was similar to the one used in the 2019 study. Namely, the threshold for dentinal lesions ($d_3/D_3$) used in the 2012 study was considered to be equivalent to the ICDAS grade 4

and more in 2019 [11]. For comparing prevalence (dichotomous variable) and caries indexes (continuous variable), logistic regression (for prevalence) and Mann-Whitney tests as well as zero-inflated negative binomial (ZINB) models (for caries experience) were used. Values were estimated for 2012 and 2019 independently at the territorial level and by province. The caries indices were considered as counting variables (cf "explanatory analysis (2019)"). Results are given for models without random effects due to high complexity of the effects considered within the comparisons (school, investigator, time, province, index) [16]. Indeed, beforehand, sensitivity analyses were conducted to verify the impact of using mixed models -with examiner and cluster effects- on the index's comparisons.

- *Explanatory analyses (2019)*

A bivariate analysis was performed to investigate the associations between each determinant (explanatory variables) and the index $d_3t + D_3T$.

According to literature [17–19], mixed models were used, including the school and examiner parameters as random effects [20]. Indeed, due to the correlation of observations within a school, the usual tests were invalid and inter-cluster variances were inflated. Moreover, the number of examiners was high and despite the calibration process, an examiner effect was expected.

The index $d_3t + D_3T$ has a count data distribution with a large fraction of zero values (approx. 40%) and an over-dispersion (variance>mean). ZIP and ZINB models were tested to verify which one presented the best indicators (AIC, BIC) of adequacy. This pilot phase justified the use of a ZINB model to adequately describe the variable distribution [21]. The ZINB model generated two separate models; the first one consists of a subpopulation at risk of caries ("with caries": $d_3t + D_3T \neq 0$), the second is composed of children considered not to be at risk ("without caries": $d_3t + D_3T = 0$).

Then, multivariate zero-inflated negative-binomial mixed-models were performed to further understand the influence of health determinants on dental caries. Three multivariate analyses were first performed for each of the three categories of explanatory variables. The inclusion of explanatory variables was decided on the basis of their significance ($p<0.1$) in the bivariate analyses or according to their clinical relevance. Within each model, multicollinearity was tested using the variance inflation factor (VIF), and covariates with a VIF above 10 or that were identified as being highly inter-related (chi-squared test) were considered to have high multicollinearity and therefore excluded from the analysis. Then, a final model with all the variables that were significant ($p<0.05$) in the previous models was conducted. The results were expressed using odds-ratios (OR) and 95% confidence intervals for prevalence (from logistic model) and with OR and Incidence Rate Ratio (IRR) and 95% confidence intervals for caries indexes (from ZINB model). Interactions between covariates were explored and their significance in the final model was evaluated. For the first three models, interactions were not tested as the objective was to identify main explanatory variables to be integrated in the final model. For the final model, each two by two interactions were explored in univariate regression analyses. Those interactions when significant were added and thus tested one by one in the final model. Only, significant interactions in the multivariate final model were considered.

## Results

### Sample description (2019)

- *Sociodemographic status, Ethnicity & conditions of living*

Of the 488 children initially selected, 413 participated (participation rate = 84.6%). The mean age was 9.11±0.31 years. Of the sample, 47.5% were boys and 81% attended a public school.

About 12% of the children lived in the islands, 28% in the north, and 60% in the south. One-third of the children identified themselves as being multiracial, 45% Oceanian, 22% European or belonging to other communities. Only 4% of the sample did not want to answer this question. It must be noticed that 32% of the children declared living in tribes or in squats (n = 8). Tables 1 and S2 respectively describe the population characteristics depending on the province and ethnicity.

- *Access to oral health care and prevention*

Of the children included, 87% declared having participated in the sealant preventive interventions at age 6. This proportion was significantly lower in the islands (66%) as compared to the north and south provinces (approx. 90%). Nearly half (47%) of the children reported brushing their teeth at school. Toothbrushing at school was better deployed in the north province as compared to the south and islands. Approximatively, half the parents declared they had difficulties accessing oral health care for their child. This rate was much higher for Oceanians (55%) as compared to Europeans & others (22%) (Tables 1 and S2).

- *Oral health behaviours*

One in five children reported drinking sweetened beverages when thirsty and about one-third said they consumed a sweetened beverage at mealtime. The majority of children declared consuming sweet snacks (89.6%) or sweet drinks (83.4%) apart from meals, this proportion was higher among Oceanians and multiracial children. One-third of children did not have breakfast before school on a regular basis but more than half reported brushing their teeth twice a day (Tables 1 and S2).

- *Dental status*

Table 2 presents children' dental status. It can be noticed that 61% of the children had at least one untreated carious lesion on deciduous or permanent teeth ($d_3t + D_3T \neq 0$) with a mean number of carious untreated teeth ($d_3t + D_3T$) of 1.78±2.13. The mean number of permanent teeth affected by the carious process ($D_3MFT$) was 0.47±1.04 and 64% of the children had at least one permanent tooth effected by dental caries when initial carious lesions were considered ($D_1MFT \neq 0$).

## Comparison of dental status between 2012 and 2019

Caries prevalence on permanent teeth declined significantly in New Caledonia between 2012 and 2019. Indeed, the % of children with untreated dental caries on permanent teeth ($D_3T \neq 0$) was 28.6% in 2012 as compared to 17.7% in 2019. The percentage of children with permanent teeth affected by the carious process ($D_3MFT \neq 0$) decreased from 35.9% to 24%. This decrease was significant in the north and south provinces. In the north, the percentage of children with $D_3MFT \neq 0$ dropped from 31.1% to 13.7%. No significant change was observed for temporary teeth ($d_3ft \neq 0$ and $d_3t \neq 0$) nor for the prevalence of initial untreated dental caries on permanent teeth ($D_1T \neq 0$) (Table 3).

The mean number of permanent teeth affected by the carious process ($D_3MFT$) decreased by approximately 20% between 2012 and 2019 from 0.76 (±1.29) to 0.47 (±1.04). Significance varied much depending on the indices or the province. In the north province, ORs of 3.48 [1.64–7.38] and 5.0 [1.63–15] were respectively found, showing that children in 2019 were three to five times more likely to be caries free as compared to 2012. For temporary teeth ($d_3t$) or combined indexes ($d_3t$-$D_3T$), the number of teeth affected in children with caries significantly decreased at the territorial level as well as in the south province. Nevertheless, when

**Table 1. Description of the study population for the explanatory variables by province (2019).**

| Variables | South | | North | | Islands | | Total | | p |
|---|---|---|---|---|---|---|---|---|---|
| | N | % | N | % | N | % | N | % | |
| Sociodemographic status, Ethnicity & conditions of living | | | | | | | | | |
| **Gender** | 248 | | 117 | | 48 | | 413 | | 0.20 |
| Male | | 43.95% | | 35.85% | | 50.0% | | 47.5% | |
| Female | | 56.05% | | 46.15% | | 50.0% | | 52.5% | |
| **Province** | | | | | | | 413 | | |
| South | | | | | | | | 60.1% | |
| North | | | | | | | | 28.3% | |
| Islands | | | | | | | | 11.6% | |
| **Ethnicity** | 226 | | 115 | | 42 | | 383 | | <0.01 |
| Oceanian | | 37.6% | | 48.7% | | 73.8% | | 44.9% | |
| European/ Others | | 21.2% | | 29.6% | | 7.1% | | 22.2% | |
| Multiracial | | 41.2% | | 21.7% | | 19.1% | | 32.9% | |
| **Place of living** | 235 | | 115 | | 45 | | 395 | | <0.01 |
| Tribe/squat | | 9.4% | | 53.9% | | 97.8% | | 32.4% | |
| Town/village/isolated property | | 90.6% | | 46.1% | | 2.2% | | 67.6% | |
| **Health insurance** | 238 | | 110 | | 41 | | 389 | | <0.01 |
| Basic public insurance only | | 13.0% | | 16.4% | | 7.3% | | 13.4% | |
| State aid supplemental | | 20.2% | | 20.0% | | 68.3% | | 25.2% | |
| Private supplemental | | 66.8% | | 63.6% | | 24.4% | | 61.4% | |
| **Type of school** | 248 | | 117 | | 48 | | 413 | | <0.01 |
| Public | | 91.1% | | 83.8% | | 22.9% | | 81.1% | |
| Private | | 8.9% | | 16.2% | | 77.1% | | 18.9% | |
| **Home sanitary equipment:** | 241 | | 117 | | 44 | | 402 | | |
| all available | | 82.6% | | 61.5% | | 25.0% | | 70.1% | |
| one or more missing | | 17.4% | | 38.5% | | 75.0% | | 29.9% | |
| Access to oral health care & prevention | | | | | | | | | |
| **number of sealed molars** | 248 | | 117 | | 48 | | 413 | | <0.01 |
| None | | 29.0% | | 48.7% | | 72.9% | | 39.7% | |
| 1–3 | | 43.2% | | 35.1% | | 20.8% | | 38.3% | |
| 4 | | 27.8% | | 16.2% | | 6.3% | | 22% | |
| **Tooth brushing at school** | 247 | | 117 | | 48 | | 412 | | <0.01 |
| Yes | | 37.25% | | 74.4% | | 29.2% | | 46.8% | |
| No | | 62.75% | | 25.6% | | 70.8% | | 53.2% | |
| **Dental attendance** | 235 | | 108 | | 35 | | 378 | | 0.798 |
| Never have visited | | 14.9% | | 17.6% | | 17.1% | | 15.9% | |
| Already have visited a dentist | | 85.1% | | 82.4% | | 78,7% | | 84.1% | |
| Visit to the dentist every year | | 12.9% | | 6.8% | | 4.2% | | 10.2% | |
| **Access to oral health care** | 242 | | 117 | | 44 | | 403 | | 0.21 |
| No difficulties | | 57.0% | | 52.1% | | 43.2% | | 54.1% | |
| Perception of difficulties | | 43.0% | | 47.9% | | 56.8% | | 45.9% | |
| **Participation to the OHP program** | 229 | | 107 | | 41 | | | | <0.01 |
| Yes | | 89.5% | | 89.7% | | 65.85% | 377 | 87% | |
| No | | 10.5% | | 10.3% | | 34.15% | | 13% | |
| Oral health behaviours | | | | | | | | | |
| **Frequency of tooth brushing** | 247 | | 116 | | 48 | | 411 | | <0.01 |

*(Continued)*

**Table 1.** (Continued)

| Variables | South | | North | | Islands | | Total | | p |
|---|---|---|---|---|---|---|---|---|---|
| | N | % | N | % | N | % | N | % | |
| Twice a day or more | | 59.9% | | 60.3% | | 29.2% | | 56.45% | |
| Once a day or less | | 40.1% | | 39.6% | | 70.8% | | 43.55% | |
| **Usual drink when thirsty** | 247 | | 117 | | 48 | | 412 | | 0.44 |
| Sweet drink/Milk | | 22.7% | | 17.1% | | 18.75% | | 20.6% | |
| Water | | 77.3% | | 82.9% | | 81.25% | | 97.4% | |
| **Usual drink during mealtime** | 246 | | 117 | | 48 | | 411 | | 0.02 |
| Sweet drink/Milk | | 37.8% | | 23.9% | | 27.1% | | 32.6% | |
| Water | | 62.2% | | 76.1% | | 72.9% | | 67.4% | |
| **Sweet drinks during weekdays** | 242 | | 117 | | 44 | | 403 | | 0.40 |
| Never | | 17.3% | | 15.4% | | 15.9% | | 16.6% | |
| Some days | | 76.9% | | 78.6% | | 70.5% | | 76.7% | |
| Daily | | 5.8 | | 6.0% | | 13.6% | | 6.7% | |
| **Sweet foods during weekdays** | 242 | | 117 | | 44 | | 403 | | 0.60 |
| Never | | 9.9% | | 11.1% | | 11.4% | | 10.4% | |
| Some days | | 86.0% | | 87.2% | | 81.8% | | 85.9% | |
| Daily | | 4.1% | | 1.7% | | 6.8% | | 3.7% | |
| **Breakfast on weekdays** | 241 | | 117 | | 44 | | 402 | | 0.12 |
| No/ some days | | 35.7% | | 22.2% | | 31.8% | | 31.3% | |
| Everyday | | 64.3% | | 77.8% | | 68.2% | | 68.7% | |

N/%; Number and percentages, p: p value for bi variate analyses.

**Table 2. Dental status of the study population (2019).**

| Dental status* | n = 413 |
|---|---|
| Number of temporary teeth (t) | 7.7±4.39 |
| Number of permanent teeth (T) | 15.73±4.41 |
| $d_3t$ | 1.47±1.83 |
| $d_3ft$ | 2.04±2.08 |
| % $d_3t$ = 0 (95%CI) | 182 *(44.1%)* [0.51–0.61] |
| % $d_3ft$ = 0 (95%CI) | 135 *(32.7%)* [0.62–0.72] |
| $d_3t + D_3T$ | 1.78±2.13 |
| % with $d_3t + D_3T$ = 0 (95%CI) | 161 *(38.98%)* [0.56–0.66] |
| $D_3T$ | 0.31±0.84 |
| % with $D_3T$ = 0 (95%CI) | 340 *(82.3%)* [0.14–0.22] |
| % with $D_1T$ = 0 (95%CI) | 158 *(38.3%)* [0.57–0.66] |
| $D_1T$ | 1.44±1.64 |
| $D_1MFT$ | 1.92±1.95 |
| $D_3MFT$ | 0.47±1.04 |
| % with $D_3MFT$ = 0 (95%CI) | 312 *(75.54%)* [0.2–0.29] |
| % with $D_1MFT$ = 0 (95%CI) | *149 (36.1%)* [0.59–0.68] |

*Mean and SD or percentages with 95% confidence interval.

DMFT: Number of Decayed (D), Missing (M) and Filled (F) Teeth (T) for permanent teeth.

dft: Number of decayed (d) and filled (f) teeth (t) for temporary teeth.

Caries detection threshold D1: ICDAS stages 1 and more, D3: ICDAS stages 4 and more.

**Table 3. Changes in prevalence of dental caries between 2012 (n = 789) and 2019 (n = 413).**

| Prevalence (% with) | South | | | North | | | Islands | | | Total | | |
|---|---|---|---|---|---|---|---|---|---|---|---|---|
| | N (%) | OR [IC] | p | N (%) | OR [IC] | p | N (%) | OR [IC] | p | N (%) | OR [IC] | p |
| **$D_3T \neq 0$** | | | | | | | | | | | | |
| 2012 | 161 (29.7%) | | | 39 (23.35%) | | | 26 (32.5%) | | | 226 (28.6%) | | |
| 2019 | 49 (19.8%) | 0.58 [0.41–0.84] | 0.004 | 13 (11.1%) | 0.41 [0.21–0.81] | 0.01 | 11 (22.9%) | 0.62 [0.27–1.40] | 0.25 | 73 (17.7%) | 0.53 [0.4–0.72] | <0.001 |
| **$d_3t \neq 0$** | | | | | | | | | | | | |
| 2012 | 265 (48.9%) | | | 96 (57.5%) | | | 60 (75.0%) | | | 421 (53.4%) | | |
| 2019 | 138 (55.7%) | 1.31 [0.97–1.77] | 0.08 | 63 (53.85%) | 0.86 [0.54–1.39] | 0.54 | 30 (62.5%) | 0.93 [0.41–2.14] | 0.87 | 231 (55.9%) | 1.11 [0.87–1.41] | 0.39 |
| **$d_3t + D_3T \neq 0$** | | | | | | | | | | | | |
| 2012 | 300 (55.4%) | | | 109 (65.3%) | | | 64 (80%) | | | 473 (59.9%) | | |
| 2019 | 154 (62.1%) | 1.32 [0.97–1.80] | 0.07 | 66 (56.4%) | 0.68 [0.42–1.11] | 0.13 | 32 (66.7%) | 0.5 [0.22–1.13] | 0.09 | 252 (61.0%) | 1.05 [0.82–1.33] | 0.72 |
| **$D_1T \neq 0$** | | | | | | | | | | | | |
| 2012 | 266 (49.1%) | | | 106 (63.5%) | | | 59 (73.7%) | | | 431 (54.6%) | | |
| 2019 | 133 (53.6%) | 1.2 [0.89–1.62] | 0.23 | 66 (56.4%) | 0.74 [0.46–1.21] | 0.23 | 36 (75.0%) | 1.07 [0.47–2.43] | 0.88 | 235 (56.9%) | 1.10 [0.86–1.39] | 0.45 |
| **$DMFT \neq 0$** | | | | | | | | | | | | |
| 2012 | 203 (37.5%) | | | 52 (31.1%) | | | 28 (35.0%) | | | 283 (35.9%) | | |
| 2019 | 69 (27.8%) | 0.64 [0.46–0.89] | 0.008 | 16 (13.68%) | 0.35 [0.19–0.65] | 0.001 | 16 (33.3%) | 0.92 [0.44–1.98] | 0.85 | 101 (24.5%) | 0.58 [0.44–0.76] | <0.001 |
| **$d_3ft \neq 0$** | | | | | | | | | | | | |
| 2012 | 332 (61.3%) | | | 116 (69.5%) | | | 61 (76.3%) | | | 509 (64.5%) | | |
| 2019 | 167 (67.3%) | 1.30 [0.95–1.79] | 0.10 | 75 (64.1%) | 0.79 [0.47–1.29] | 0.34 | 36 (75.0%) | 0.93 [0.41–2.14] | 0.87 | 278 (67.3%) | 1.13 [0.88–1.45] | 0.97 |
| **$D_1MFT \neq 0$** | | | | | | | | | | | | |
| 2012 | 337 (62.2%) | | | 125 (74.85%) | | | 64 (80.0%) | | | 526 (66.7%) | | |
| 2019 | 151 (60.9%) | 0.95 [0.69–1.29] | 0.73 | 71 (60.7%) | 0.52 [0.31–0.86] | 0.01 | 42 (87.5%) | 1.75 [0.63–4.83] | 0.28 | 264 (63.9%) | 0.88 [0.69–1.13] | 0.34 |
| **$df_3t + D_3MFT \neq 0$** | | | | | | | | | | | | |
| 2012 | 370 (68.3%) | | | 131 (78.4%) | | | 66 (82.5%) | | | 567 (71.9%) | | |
| 2019 | 185 (74.6%) | 1.36 [0.97–1.91] | 0.07 | 78 (66.67%) | 0.55 [0.32–0.94] | 0.03 | 39 (81.3%) | 0.92 [0.36–2.32] | 0.86 | 302 (73.1%) | 1.06 [0.82–1.39] | 0.64 |

p: p values for logistic regression model without random effect.

N: Number (% percentage).

OR [IC] Odd Ratio with 95% Confidence Interval.

DMFT: Number of Decayed (D), Missing (M) and Filled (F) Teeth (T) for permanent teeth.

dft: Number of decayed (d) and filled (f) teeth (t) for temporary teeth.

Caries detection threshold / D1: ICDAS stages 1 and more, D3: ICDAS stages 4 and more.

initial carious lesions (stage $D_1$) were counted on permanent teeth, there was an increase in caries experience at the territorial level and in the north province (S3 and S4 Tables).

## Explanatory analyses (2019)

### • *Bivariate analyses*

Mixed models showed that among children with untreated carious lesions ($d_3t+D_3t\neq0$), the mean number of carious teeth ($d_3t+D_3t$) was associated to the following variables: province, ethnicity, number of sealed molars, access to oral health care, toothbrushing frequency or use of sweet foods ($p<0.05$). As an example, in this group, the risk for having untreated carious lesions was higher for Oceanian or multiracial children. For "caries-free" children, caries status was significantly related to access to oral health care ($p<0.05$) (S5 Table).

### • *Multivariate analyses*

The first multivariate analysis (Model 1: social determinants) found a significant influence of ethnicity with Oceanian and multiracial children being more at risk of caries with Incident Rate Ratios of 2.06 and 1.76 respectively as compared to "Europeans & Others" (Table 4). The number of sealed molars and difficulties in accessing oral health care were associated to the number of untreated dental caries in model 2 (oral health care model) (Table 5). In the behavioural model (Model 3), collinearity was found between the diet related variables; the variable "frequency of consumption of sweet snacks during weekdays" was the only one kept in model 3. In this model, the toothbrushing frequency and frequent consumption of sweet snacks were identified as risk factors for being in the "with caries" group (Table 6). In the overall model (Table 7), Oceanian children, with no sealed molars, whose parents declared difficulty in accessing oral health care, who had frequent sugary snacks had a significantly higher probability of untreated dental caries. Toothbrushing frequency was almost significant. On the other

**Table 4. Multivariate analysis: Relationship between sociodemographic variables and the number of untreated carious lesions ($d_3t+D_3t$) in 2019 (n = 382).**

| Sociodemographic status, Ethnicity & living conditions | Multivariate analysis ($d_3t+D_3t$) | | | | | |
|---|---|---|---|---|---|---|
| | Zero-inflated (Without caries) | | | Negative binomial (with caries) | | |
| | OR | 95% CI | p | IRR | 95% CI | p |
| **Gender** | | | | | | |
| Male *vs* female | 1.24 | [0.48–3.19] | 0.66 | 0.86 | [0.66–1.11] | 0.25 |
| **Province** | | | | | | |
| South [Ref] | | | | | | |
| North | 3.77 | [0.85–16] | 0.08 | 1.33 | [0.96–1.85] | 0.09 |
| Islands | 2.50 | [0.3–21] | 0.39 | 1.06 | [0.68–1.67] | 0.79 |
| **Ethnicity** | | | | | | |
| European &Others [Ref] | | | | | | |
| Oceanian | 0.45 | [0.09–2.22] | 0.33 | 2.06 | [1.36–3.11] | <0.001 |
| multiracial | 1.51 | [0.40–5.77] | 0.54 | 1.76 | [1.13–2.75] | <0.01 |
| **Sanitary equipment** | | | | | | |
| >1 missing vs all available | 0.32 | [0.08–1.21] | 0.09 | 1.12 | [0.83–1.51] | 0.45 |

OR = odds ratio, IRR: Incidence rate ratio, 95%CI: 95% Confidence Interval.

p: p value for + Zero-inflated negative binomial regression with random effects.

Ref: Reference for the comparison.

Dispersion parameter = 1,87.

**Table 5. Multivariate analysis: Relationship between access to oral health care variables and the number of untreated carious lesions ($d_3t+D_3t$) in 2019 (n = 368).**

| Access to oral health care & prevention | Multivariate analysis ($d_3t+D_3t$) | | | | | |
|---|---|---|---|---|---|---|
| | Zero-inflated (Without caries) | | | Negative binomial (with caries) | | |
| | OR | 95% CI | p | IRR | 95% CI | p |
| **Number of sealed molars** | | | | | | |
| None vs At least one | 2.21 | [0.88–5.59] | 0.09 | 1.5 | [1.13–1.99] | <0.01 |
| **Tooth brushing at school** | | | | | | |
| Yes vs No | 1.01 | [0.41–2.47] | 0.98 | 1.09 | [0.82–1.45] | 0.56 |
| **Participation to the OHP program** | | | | | | |
| No vs Yes | 0.59 | [0.16–2.21] | 0.44 | 0.87 | [0.59–1.28 | 0.48 |
| **Access to oral health care** | | | | | | |
| Difficulties vs No difficulties | 0.18 | [0.07–0.51] | <0.01 | 1.54 | [1.17–2.03] | <0.01 |

OR = odds ratio, IRR: Incidence rate ratio, 95%CI: 95% Confidence Interval.

p: p value for + Zero-inflated negative binomial regression with random effects.

Dispersion parameter = 2,62.

hand, children who had no difficulty accessing dental care were more likely to be in the cavity-free group. For the variables included in the overall model, some interactions were found in univariate analyses but none was significant in the multivariate analysis, they were thus not added in the model (Table 7).

## Discussion

This study provides an overview of changes in the prevalence and experience of dental caries in 9-years-old schoolchildren in New Caledonia between 2012 and 2019. It showed that the epidemiological situation has improved after five years of implementation of the oral health promotion program. However, results indicate that changes varied depending on the province, caries detection threshold and whether or not temporary teeth were included. Overall, dental status improved in the north and slightly in the south province while no significant change was observed in the islands. Dental status for permanent teeth in New Caledonia in 2019 was found to be much better as compared to values observed in high income countries according

**Table 6. Multivariate analysis: Relationship between behaviours & lifestyles variables and the number of untreated carious lesions ($d_3t+D_3t$) in 2019 (n = 400).**

| Oral health behaviours | Multivariate analysis ($d_3t+D_3t$) | | | | | |
|---|---|---|---|---|---|---|
| | Zero-inflated (Without caries) | | | Negative binomial (with caries) | | |
| | OR | 95% CI | p | IRR | 95% CI | p |
| **Frequency of tooth brushing** | | | | | | |
| Less than twice a day vs Twice a day | 0.54 | [0.24–1.24] | 0.15 | 1.37 | [1.05–1.8] | 0.02 |
| **Frequency of consumption of sweet foods during weekdays** | | | | | | |
| Some days /Every day Vs No day | 2.13 | [0.23–19] | 0.5 | 1.37 | [1.05–1.78] | 0.01 |
| **Frequency of taking breakfast on weekdays before school** | | | | | | |
| No day/ Some days vs Every day/daily | 0.3 | [0.07–1.16] | 0.08 | 1.15 | [0.87–1.5] | 0.33 |

OR = odds ratio, IRR: Incidence rate ratio, 95%CI: 95% Confidence Interval.

p: p value for Zero-inflated negative binomial regression with random effects.

Dispersion parameter: 2,69.

Table 7. Final multivariate analysis: Health determinants and risk factors associated with untreated carious lesions ($d_3t+D_3t$) in 2019 (n = 381).

| All type of variables | Multivariate analysis ($d_3t+D_3t$) | | | | | |
| --- | --- | --- | --- | --- | --- | --- |
| | Zero-inflated (Without caries) | | | Negative binomial (with caries) | | |
| | OR | 95% CI | p | IRR | 95% CI | p |
| **Ethnicity** | | | | | | |
| European/ Others [Ref] | | | | | | |
| Oceanian | 0.56 | [0.13–2.39] | 0.44 | 1.58 | [1.03–2.44] | 0.03 |
| multiracial | 1.62 | [0.43–6.05] | 0.47 | 1.37 | [0.87–2.15] | 0.17 |
| **Number of sealed molars** | | | | | | |
| None *vs* at least one | 2.06 | [0.76–5.62] | 0.16 | 1.31 | [1.01–1.70] | 0.03 |
| **Access to oral health care** | | | | | | |
| Difficulties *vs* No difficulties | 0.12 | [0.03–0.49] | <0.01 | 1.36 | [1.04–1.78] | 0.02 |
| **Frequency of consumption of sweet foods during weekdays** | | | | | | |
| Some days /Every day Vs No day | 2.75 | [0.35–21] | 0.33 | 1.69 | [1.07–2.66] | 0.02 |
| **Frequency of tooth brushing** | | | | | | |
| Less than twice a day vs Twice a day | 0.42 | [0.14–1.24] | 0.11 | 1.29 | [1.00–1.69] | 0.06 |

OR = odds ratio, IRR: Incidence rate ratio, 95%CI: 95% Confidence Interval.

p: p value for Zero-inflated negative binomial regression with random effects.

Ref: Reference for the comparison.

Dispersion parameter: 3,55.

to the WHO Oral health report (prevalence 17,7% as compared to 29,3% in the report). Nevertheless, the situation in New Caledonia appeared to be less favourable when deciduous teeth were considered with a prevalence of carious lesions of 56% as compared to 38% in the report [22]. The study also identified the risk factors and social determinants of children's oral health status in 2019. Results showed that being Oceanian, having difficulties in accessing oral health care and unfavourable health behaviours were associated with being in the "with caries" group.

Several strengths can be underlined. First, the context is original as the study is linked to the measurement of the health impact of the program "My Teeth My Health" extended progressively since 2014 across New Caledonia. This program involves not only health professionals but also local institutions, social welfare organizations and the educational community taking advantage of the school environment to create a supportive environment for health [23]. As an example, following the implementation of the program, 65% of the schools declared in 2018, having implemented toothbrushing at school [23]. Second, the study is methodologically sound, similar to the 2012 study. It was conducted upon a representative sample that allowed encompassing the cultural, geographic, and social diversity of New Caledonia. Another strength was the use of zero-inflated negative binomial models that account for the excess of zeros and over-dispersion of caries experience indexes [21]. It must also be noticed that mixed models were used for the explanatory analyses that integrated the cluster and investigator effects, lowering the risk of bias for all variables simultaneously [20]. Third, the choice of the 9-year-old age group allowed to evaluate dental status of children who have been receiving oral health promotion interventions for approximately 3 to 4 years since age 5–6 years (tooth brushing at school, molars sealed). Another feature of this research is that analyses were intended to obtain a better understanding of oral health inequalities, namely by analysing determinants such as ethnicity, geographical and socioeconomic factors. Taking these determinants into account is indeed essential to be able to understand the distribution of oral diseases in New Caledonian children [2].

Some limitations must be mentioned. First, results allowed to identify correlational links between the participation in the oral health promotion program and the 2019 dental status. However, the methodology used does not permit to conclude with causal relationships [23]. Indeed, there is limited knowledge of the oral health promotion program implementation conditions and its deployment within the provinces apart from some qualitative information available from the health agency [12,24]. Moreover, the program integrates multiple and complex actions which impacts cannot be identified with two successive descriptive studies, without a comparison group [25,26]. It must also be noticed that many explanatory variables were collected through child self-administered questionnaires based on declarative and subjective perceptions. It is therefore likely that unfavourable health habits such as infrequent toothbrushing were under-estimated. Another limitation is the change of diagnostic criteria between 2012 and 2019. In 2012, the Ekstrand classification was used; the presence of a cavity determines the transition from $D_1$ (non cavitated stage) to $D_3$ (cavitated stage), with an underestimation of non-cavitated dentinal lesions [27]. In 2019, the ICDAS classification was used that allows better detection of non cavitated lesions [15]. To allow comparisons, the threshold "$D_1$" was associated to ICDAS stages 1 to 3. This might have led to a measurement bias that could explain the increase in the number of early carious lesions ($D_1$) observed between 2012 and 2019. However, there were not many differences between 2012 and 2019 in the criteria used for the detection of more advanced lesions, which facilitated the comparisons for indexes based on the D3 threshold.

Explanatory analysis revealed that dental status was related to children' oral health behaviours and access to oral health care but also to ethnicity with Oceanian children being more are risk for oral diseases [28–30]. Indeed, Oceanian children had specific social characteristics as nearly half of them lived in a tribe, lacking sanitary facilities more frequently. Those characteristics are in line with the situation of the Kanak community (Oceanian) that has a customary lifestyle and is highly affected by poverty. In 2019, the median standard of living of non-Kanak households (234,200 CFP francs) was more than twice that of Kanak households (116,800 CFP francs) [31]. To better understand those gaps, it is essential to mention the territory's colonial history that resulted in deep inequalities even if since 1986 (The Noumea treaty) a process of reconciliation and political autonomy is occurring [32,33]. Our results are in line with previous studies conducted in Canada [34], the United States [35,36], Australia [37] and New Zealand [38] and with a systematic review [39] showing that both children and adults from Indigenous communities are at higher risk of dental caries. Moreover, studies conducted in the pacific region, including New Caledonia, have reported ethnic disparities for other chronic diseases [40] and cancers [41]. In New Caledonia, Oceanian children are more frequently affected by obesity [42–44] and rheumatic heart diseases [44]. In addition to possible genetic influence, socio-environmental factors have been implicated [41]. Thus, indigenous populations can be identified as a priority target group for oral health promotion interventions such as in the Australian Oral Health Plan (2015–2024) [45].

The concepts of social exclusion and intersectionality are key aspects explaining health inequities in New Caledonia. Social exclusion is a multi-dimensional, dynamic and relational concept that affects people's experience and intensity of marginalization [46,47]. This process integrates different aspects of fragility, including social, economic, cultural and political factors but also time, place and context [5]. The notion of intersectionality provides a better understanding of the complexity of social exclusion. This concept recognizes that people who suffer from exclusion face multiple forms of discrimination related to age, gender, race or ethnicity that are neither isolated nor independent. Factors of disadvantage intersect, and layers of discrimination accumulate, exacerbating social exclusion [48]. Thus, people who experience multiple social exclusions build up negative experiences from childhood that exclude them from

social interactions, education, work and health services leading to extreme health conditions [5]. In 2015, a qualitative study assessed the barriers and levers to dental care accessibility for children in New Caledonia [49]. Results identified external barriers such as the complexity of the health system, the low density of dentists and low number of public structures. But some other barriers were linked to parents' curative vision of dental care and their negative opinions of dental practitioners or past negative experiences with dental care. Thus, when linking those two studies, the social interaction between dental practitioners and families belonging to the indigenous community can be questioned and namely how this interaction can affect children's oral health. A socio-cultural gap may exist between dentists and Oceanian patients regarding financial capacity, health literacy and cultural representations of healthcare. In addition, judgments from both sides towards each other might worsen the situation, with on one hand, dentists considering patients with poor oral health as having a chaotic lifestyle and on the other hand, patients discussing the pertinence of dental treatments being offered. Such negative interactions bring the "us versus them" or "othering" concept, reinforcing social exclusion within the dental care environment. Thus, measuring the effects of health promotion interventions on oral health considering health inequities and conditions of implementation is very important. Indeed, health promotion interventions can be effective for the whole population with nonetheless an increase in health inequities when the positive effects are lowered in children in deprived areas with higher health need.

The evolution of children's caries status between 2012 and 2019 was studied by province. Indeed, the oral health promotion program is mainly managed at the province level where the health initiatives and mobilization of local actors may vary leading to different dynamics in the implementation of the program and thus resulting in different evolutions of children's oral health. It should be remembered that the north province and the islands are two disadvantaged areas as compared to the south, with high poverty rates and a large percentage of indigenous population. The significant improvement of dental status in the northern province could be related to the success of the implementation of the program in this area with probably high stakeholder's involvement [50]. In the future, qualitative studies should be conducted to complement existing quantitative epidemiological data [51]. The combined analysis of quantitative and qualitative results (mixed methods) would provide solutions to improve the program [50]. Namely, it would be useful to explore the conditions of interventions in the north province in order to be able to build on their experiences and to better understand the barriers encountered in the islands. Indeed, this study allowed to appreciate the potential oral health impacts of the program. The evaluation of the program that is being conducted by the health agency will bring soon more information about its pertinency and the quality of interventions, its strengths and limitations as well as the variations in the conditions of implementation namely by province. Unfortunately, the evaluation has been delayed due to the COVID 19 pandemic.

This study showed that dental status improved overall in New Caledonia since 2012. This improvement might be linked to the implementation of the oral health promotion program, particularly in the northern province where the conditions for implementation were favorable. The study also showed that health inequalities namely related to ethnicity persisted, highlighting the complexity of the determinants of oral health and the importance of monitoring the issue of health inequities. In the future, it would also be useful to conduct other epidemiological studies on a regular basis, particularly to evaluate the impact of the COVID-19 crisis [52]. Indeed, the territory remained closed for 2 years with an interruption of non-essential preventive activities and that may have negatively impacted children's oral health. The health agency will also have to adapt the oral health promotion program in terms of content and priorities in view of the present results.

## Supporting information

**S1 Table. Description of the study explanatory variables.**
(DOCX)

**S2 Table. Description of the study population for the explanatory variables by ethnicity.**
(DOCX)

**S3 Table. Comparison of caries indexes between 2012 and 2019 –non parametric test.**
(DOCX)

**S4 Table. Comparison of caries indexes between 2012 and 2019—ZINB models.**
(DOCX)

**S5 Table. Bivariate analysis: Relationship between caries experience and explanatory variables.**
(DOCX)

## Acknowledgments

The authors would like to express our gratitude to Caroline Eschevins (Université Clermont Auvergne, CROC, F-63000, Clermont-Ferrand, France) for her help in the writing of the manuscript. We would like also to thank the school teaching teams and administrative staffs for their collaboration and support.

## Author Contributions

**Conceptualization:** Helene Pichot, Stephanie Tubert-Jeannin.

**Data curation:** Amal Skandrani, Bruno Pereira.

**Formal analysis:** Amal Skandrani, Bruno Pereira.

**Investigation:** Helene Pichot.

**Methodology:** Helene Pichot, Bruno Pereira.

**Project administration:** Helene Pichot.

**Resources:** Helene Pichot.

**Software:** Bruno Pereira.

**Supervision:** Stephanie Tubert-Jeannin.

**Writing – original draft:** Amal Skandrani, Stephanie Tubert-Jeannin.

**Writing – review & editing:** Amal Skandrani, Estelle Pegon-Machat, Stephanie Tubert-Jeannin.

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
