## [Decision Letter · Decision Letter 0]

14 Nov 2022

PONE-D-22-27517Measuring impacts of oral health promotion interventions on health inequities: the example of New CaledoniaPLOS ONE

Dear Dr. tubert-jeannin,

Thank you for submitting your manuscript to PLOS ONE. After careful consideration, we feel that it has merit but does not fully meet PLOS ONE’s publication criteria as it currently stands. Therefore, we invite you to submit a revised version of the manuscript that addresses the points raised during the review process.

We look forward to receiving your revised manuscript.

Kind regards,

César Félix Cayo-Rojas, Ph.D.

Academic Editor

PLOS ONE

Journal Requirements:

Reviewers' comments:

Reviewer's Responses to Questions

**Comments to the Author**

1. Is the manuscript technically sound, and do the data support the conclusions?

Reviewer #1: Partly

Reviewer #2: Partly

2. Has the statistical analysis been performed appropriately and rigorously? 

Reviewer #1: Yes

Reviewer #2: Yes

3. Have the authors made all data underlying the findings in their manuscript fully available?

Reviewer #1: Yes

Reviewer #2: Yes

4. Is the manuscript presented in an intelligible fashion and written in standard English?

Reviewer #1: Yes

Reviewer #2: Yes

5. Review Comments to the Author

Reviewer #1: In materials and methods, it should include the inclusion and exclusion criteria of the study sample.

In the data collection section of the study variable, could you clarify what were the criteria and/or indicator and the procedure used to record sealants in permanent molars?

In the discussion, the importance of the study should be argued, that is, why it is important to measure the impact of health promotion interventions on oral health considering health inequities

Statistical analysis has been adequately performed, findings data are available, and the manuscript is written in standard English.

Reviewer #2: Congratulations to the team for the great work, it has very good results and contributes a lot to the study of inequalities.

Here are some comments:

Line 1. The impact evaluation implies having a real comparison group or a counterfactual and a clear health outcome in which they will evaluate the "impact", for how the article is I suggest to review it carefully.

Line 36. Review the use of abbreviations throughout the manuscript for consistency.

Line 89. Why only 9 years old group?

Line 98. It is not clear the level of representativeness, is it national? only for NC? or is it part of a national survey where one of the provinces where data was collected is NC?

Line 104-108. How much was the prevalence and power? Did you compute the ICC? If so, how much is it?

Line 109. Which rules-of-thumbs?

Line 105 vs line 111. It's a little confusing that they put the required sample size and then the sufficient sample size.

Line 129. You do not say before what ICDAS is

Line 134. And not the adult responsible for the child) (who are not necessarily the parents)

Line 136. Province of residence or birth?

Line 149. What kind of errors or biases?

Line 153. You did not previously describe the severity

Line 160. “Complete” means all age groups?

Line 162. ICC was consider in the models?

Line 187. Did you consider using interaction terms or focusing the variables?

Line 202. Table 1. Use Total instead of “NC”, or the complete name.

• Health insurance: What type of services or care the first two categories cover.

• Dental attendance: How many is “high”?

• Access to oral health: What mean “no difficulties”

Line 221. Are you sure that mean and SD are the best summary statistics? (I guess median and IQR) please, evaluate the distribution.

Line 224. Table 2

• I strongly suggest not to use abbreviations for variables, better to put them at the foot of the table.

• Are you sure that mean and SD are the best summary statistics

Line 250. Why did you fraction the models?

Line 320. Is a strong limitation for the comparison of the two years, I suggest explaining it better.

Would expect more discussion on the impact of oral health programs in the province, country or region.

Review whether they are actually doing an evaluation or measurement of impact.

Suggest putting in context how the province is doing relative to the country, suggest using disease burden articles:

https://www.scielo.br/j/rsbmt/a/rvpDy9npBdFcgM89TtgLrCr/abstract/?lang=en

https://journals.sagepub.com/doi/abs/10.1177/0022034519894963

https://journals.sagepub.com/doi/abs/10.1177/0022034517693566

6. PLOS authors have the option to publish the peer review history of their article (what does this mean?). If published, this will include your full peer review and any attached files.

Reviewer #1: No

Reviewer #2: No

---

## [Author Response · Author response to Decision Letter 0]

28 Dec 2022

The answers to the reviewer comments are given in the file " response to reviewers"

---

## [Decision Letter · Decision Letter 1]

20 Feb 2023

PONE-D-22-27517R1Influence of an oral health promotion program on the evolution of dental status in New Caledonia : a focus on health inequities.PLOS ONE

Dear Dr. Stephanie Tubert-Jeannin,

Thank you for submitting your manuscript to PLOS ONE. After careful consideration, we feel that it has merit but does not fully meet PLOS ONE’s publication criteria as it currently stands. Therefore, we invite you to submit a revised version of the manuscript that addresses the points raised during the review process.

ACADEMIC EDITOR:

I hope this mail finds you in good health. Please respond to all remarks point by point. You should improve the presentation of your results and clarify the methodology.

If you disagree on any point, then you should substantiate why.

Kind regards

We look forward to receiving your revised manuscript.

Kind regards,

César Félix Cayo-Rojas, Ph.D.

Academic Editor

PLOS ONE

Reviewers' comments:

Reviewer's Responses to Questions

**Comments to the Author**

1. If the authors have adequately addressed your comments raised in a previous round of review and you feel that this manuscript is now acceptable for publication, you may indicate that here to bypass the “Comments to the Author” section, enter your conflict of interest statement in the “Confidential to Editor” section, and submit your "Accept" recommendation.

Reviewer #1: (No Response)

Reviewer #2: (No Response)

2. Is the manuscript technically sound, and do the data support the conclusions?

Reviewer #1: Partly

Reviewer #2: Partly

3. Has the statistical analysis been performed appropriately and rigorously? 

Reviewer #1: Yes

Reviewer #2: N/A

4. Have the authors made all data underlying the findings in their manuscript fully available?

Reviewer #1: Yes

Reviewer #2: No

5. Is the manuscript presented in an intelligible fashion and written in standard English?

Reviewer #1: Yes

Reviewer #2: Yes

6. Review Comments to the Author

Reviewer #1: The following observations were made:

In materials and methods it should include the inclusion and exclusion criteria of the study sample.

In the data collection section of the study variable, could you clarify what were the criteria and/or indicator and the procedure used for the registration of sealants in permanent molars?

In the discussion, the importance of the study should be argued, that is, because it is important to measure the impact of health promotion interventions on oral health considering health inequities.

However, the manuscript does not observe the removal of these observations. If in case they have consigned it, please could you highlight it in order to verify the changes.

Reviewer #2: It is a great paper, it has improved considerably, it has a great potential of information for decision making. However, I share some comments.

Abstract

It is not clear whether the background was conducted in NC or the country.

In methods do not present ZINB model

For model 1, 3 and 4 it is not clear what the dependent variable is.

I suggest improving the conclusion

Population and Methods

Study population: What happened with the 2012 group, was it not also analyzed for comparison?

Study variables and data collection: I do not identify the operational definition of severity.

Statistical analyses: What do you mean by duplicate, what errors do you mean and how were they corrected?

Comparison of dental status between 2012 and 2019: Regarding severity, because of how it is measured, is it not an ordinal variable? if so, the model would be different from what they propose.

o Severity is a count variable?

For ZINB model, how did you assess overdispersion?

It is not clear how the comparison was between the two years, was it an adjustment variable (year) or were they estimating prevalences for each year independently?

Explanatory analyses (2019): If you found multicollinearity, did you explore making transformations in the variables?

If interactions were significant, how much variation was there in the ORs of the variables of interest with and without interaction?

If you consider that there are many explanatory variables, why did you not develop a composite index?

Results:

For "Table 2", abbreviations in the rows follow, please improve the table. Indicate the year in all tables

Table 3, without legend of what the abbreviations mean.

Is the measurement and assessment of inequalities based on comparisons between regions?

7. PLOS authors have the option to publish the peer review history of their article (what does this mean?). If published, this will include your full peer review and any attached files.

Reviewer #1: No

Reviewer #2: No

---

## [Author Response · Author response to Decision Letter 1]

5 Apr 2023

Reviewer #1: The following observations were made:

1-In materials and methods it should include the inclusion and exclusion criteria of the study sample.

The authors have fully respected, in the material and method section of the manuscript, the framework for descriptive epidemiology, namely the items listed table 1 relating to participants ( target population, sampling method, analytical sample) in " Catherine R Lesko, Matthew P Fox, Jessie K Edwards, A Framework for Descriptive Epidemiology, American Journal of Epidemiology, Volume 191, Issue 12, December 2022, Pages 2063–2070, https://doi.org/10.1093/aje/kwac115"

2 In the data collection section of the study variable, could you clarify what were the criteria and/or indicator and the procedure used for the registration of sealants in permanent molars?

The criteria, procedure and indicators used for the registration of sealants in permanent molars is now mentioned in the data collection section.

3 In the discussion, the importance of the study should be argued, that is, because it is important to measure the impact of health promotion interventions on oral health considering health inequities. However, the manuscript does not observe the removal of these observations. If in case they have consigned it, please could you highlight it in order to verify the changes.

In the discussion, this aspect is now highlighted as follow “This study showed that dental status improved overall in New Caledonia since 2012. This improvement might be linked to the implementation of the oral health promotion program, particularly in the northern region where the conditions for implementation were favorable. The study also showed that health inequalities namely related to ethnicity persisted, highlighting the complexity of the determinants of oral health and the importance of monitoring the issue of health inequities.

Reviewer #2: It is a great paper, it has improved considerably, it has a great potential of information for decision making. However, I share some comments.

Abstract

1-It is not clear whether the background was conducted in NC or the country.

The administrative status of New Caledonia has been added and it is now indicated that the 2019 study was also conducted in New Caledonia. 

2 In methods do not present ZINB model

It has been deleted

3 For model 1, 3 and 4 it is not clear what the dependent variable is.

We have added more details in the abstract to make this clearer

4 I suggest improving the conclusion

The conclusion has been rewritten

Population and Methods

5 Study population: What happened with the 2012 group, was it not also analyzed for comparison?

The material and method section has been modified so that to indicate more clearly when the text is referring to the methodology of the 2019 study and when it is referring to the comparison with the 2012 study (already published, ref 10). Moreover a short paragraph has been added to describe the sample used in 2012.

6 Study variables and data collection: I do not identify the operational definition of severity.

The word “severity” has been replaced by the words “ caries indexes” or “caries experience” in the text of the manuscript

7 Statistical analyses: What do you mean by duplicate, what errors do you mean and how were they corrected?

The study data were entered twice by two different operators using Access software. Then, the correction of data entry errors was carried out by crossing the two databases via an Excel spreadsheet. The sentence has been modified to make it clearer to the reader (A double entry was carried out and discrepancies were checked.)

8 Comparison of dental status between 2012 and 2019: Regarding severity, because of how it is measured, is it not an ordinal variable? if so, the model would be different from what they propose.Severity is a count variable?

The cumulative caries experience is calculated using DMF indices, which were considered as counting variables due to their distribution (index varying from 0 to 20- 32, skewed distribution with many 0 values). The ZINB model was therefore the more appropriate model as justified in the answers to the next comment. This is now mentioned in the manuscript. 

9 For ZINB model, how did you assess overdispersion?

Before choosing the statistical model we analysed our independent variable (D3T+ d3t): a count variable with a variance that is greater to the mean (m=1.78±2.13) and an important fraction of zero values (approx. 40%) (cf table 2). The use of the Poisson distribution (ZIP) was therefore not adequate. We then have chosen to use a negative binomial distribution (ZINB). We have nevertheless carried out the two models in order to verify which one presented the best indicators (AIC,BIC) of adequacy. Negative binomial regression was more appropriate, with lower AIC & BIC parameters, as suggested in the literature. (Cf réf 20. Preisser JS, Stamm JW, Long DL, Kincade ME. Review and recommendations for zero-inflated count regression modeling of dental caries indices in epidemiological studies. Caries Res. 2012;46:413–23.). Within the ZINB models, the alpha coefficients allowed to verify the dispersion with values approx=2. It appeared that the ZINB model thus helped to overcome overdispersion. (R Fitriani, L N Chrisdiana and A Efendi, Simulation on the Zero Inflated Negative Binomial (ZINB) to Model Overdispersed, Poisson Distributed Data. 2019 IOP Conf. Ser.: Mater. Sci. Eng. 546 052025). The values of the alpha coefficients have been added in the legend of the tables for the 4 models.

10 It is not clear how the comparison was between the two years, was it an adjustment variable (year) or were they estimating prevalences for each year independently

Prevalence & caries indexes were calculated for each year independently without adjustment for the year. This is now indicated in the text. 

11 Explanatory analyses (2019): If you found multicollinearity, did you explore making transformations in the variables? 

First, clinical pertinency but also coherence with the structure of the explanatory variables used in 2012, have driven the strategy for constructing the multidimensional models. Multicollinearity was tested using the variance inflation factor (VIF), and covariates with a VIF above 10 or that were identified as being highly inter-related (chi-squared test) were considered to have high multicollinearity. Those variables were diet variables (table 1) that were measuring almost the same thing: the frequency of consumption of sugary drinks or sugary foods (sweet drinks when thirsty, during weekdays, during mealtime). Those variables showed high collinearity. Thus we have chosen to keep only one of them. We kept the variable "Sweet foods during weekdays" as it was significant in bidimensional analysis (table S5).This is now more clearly mentioned in the text.

12 If interactions were significant, how much variation was there in the ORs of the variables of interest with and without interaction?

We have searched for interactions using a simple regression analysis. We found the interactions listed in our previous answer to you. Then we have tested one by one the influence of each interaction in the ZINB model. No interaction was found as being significant in the final model. Thus we felt it is not adequate to add a paragraph concerning the impact of these interactions. Some text has been added in the manuscript concerning interactions. 

13 If you consider that there are many explanatory variables, why did you not develop a composite index?

The objectives of the study were mainly descriptive and explicative but not predictive. The objective was to identify the determinants and risk factors explaining the presence of untreated dental caries in 2019 and to observe the evolution of the prevalence and experience of dental caries in New Caledonia since 2012. Our study was therefore not predictive. However, the reviewers' comment is relevant and interesting, our work could be completed in the future with further analyses based on the use of a composite index to predict risk factors for oral diseases. It should be noticed that the 2012 study allowed to evaluate common risk indicators for oral diseases and obesity with the use of a composite score (COGHI).

Cf Tubert-Jeannin, S., Pichot, H., Rouchon, B., Pereira, B., & Hennequin, M. (2018). Common risk indicators for oral diseases and obesity in 12-year-olds: a South Pacific cross sectional study. BMC Public Health, 18(1), 1-12.

Results:

14 For "Table 2", abbreviations in the rows follow, please improve the table. 

Improvements were made with the addition of appropriate initials

Indicate the year in all tables

Done

15 Table 3, without legend of what the abbreviations mean.

Legend has been added 

16 Is the measurement and assessment of inequalities based on comparisons between regions?

The province (region) is important for public health activities in NC. It influences the level of implementation of the program as public health structures are partly managed at a province level (cf discussion and new conclusion). It is also an ecological determinant, interlinked with ethnic distribution & social situation (cf introduction) .

---

## [Decision Letter · Decision Letter 2]

25 Apr 2023

PONE-D-22-27517R2Influence of an oral health promotion program on the evolution of dental status in New Caledonia : a focus on health inequities.PLOS ONE

Dear Dr. Stephanie Tubert-Jeannin

Thank you for submitting your manuscript to PLOS ONE. After careful consideration, we feel that it has merit but does not fully meet PLOS ONE’s publication criteria as it currently stands. Therefore, we invite you to submit a revised version of the manuscript that addresses the points raised during the review process.

ACADEMIC EDITOR:

Thank you very much for responding to all comments. I now ask you to please lift the last remarks raised by a reviewer.

We look forward to receiving your revised manuscript.

Kind regards,

César Félix Cayo-Rojas, Ph.D.

Academic Editor

PLOS ONE

Journal Requirements:

Reviewers' comments:

Reviewer's Responses to Questions

**Comments to the Author**

1. If the authors have adequately addressed your comments raised in a previous round of review and you feel that this manuscript is now acceptable for publication, you may indicate that here to bypass the “Comments to the Author” section, enter your conflict of interest statement in the “Confidential to Editor” section, and submit your "Accept" recommendation.

Reviewer #1: All comments have been addressed

Reviewer #2: (No Response)

2. Is the manuscript technically sound, and do the data support the conclusions?

Reviewer #1: (No Response)

Reviewer #2: Yes

3. Has the statistical analysis been performed appropriately and rigorously? 

Reviewer #1: (No Response)

Reviewer #2: Yes

4. Have the authors made all data underlying the findings in their manuscript fully available?

Reviewer #1: (No Response)

Reviewer #2: Yes

5. Is the manuscript presented in an intelligible fashion and written in standard English?

Reviewer #1: (No Response)

Reviewer #2: Yes

6. Review Comments to the Author

Reviewer #1: (No Response)

Reviewer #2: It is important to reinforce the justification of the interactions that were considered in the models.

Please avoid abbreviations in tables as well as in the description of results.

Reinforce the scope of public policy in oral health and how it is part of the national and global agenda.

7. PLOS authors have the option to publish the peer review history of their article (what does this mean?). If published, this will include your full peer review and any attached files.

Reviewer #1: No

Reviewer #2: No

---

## [Author Response · Author response to Decision Letter 2]

11 May 2023

Reviewer #2: 

It is important to reinforce the justification of the interactions that were considered in the models.

We have now better explained, in the material and method section, how interactions were managed in the models

Please avoid abbreviations in tables as well as in the description of results.

We have added some text in order to help non dental professionals to understand the carious indexes (well known in the dental field)

Reinforce the scope of public policy in oral health and how it is part of the national and global agenda.

A paragraph has been added in the introduction referring to the WHO resolution, thus positioning the oral health promotion program within the WHO strategy and action plan for oral health

---

## [Decision Letter · Decision Letter 3]

30 May 2023

Influence of an oral health promotion program on the evolution of dental status in New Caledonia : a focus on health inequities.

PONE-D-22-27517R3

Dear Dr. stephanie tubert-jeannin

We’re pleased to inform you that your manuscript has been judged scientifically suitable for publication and will be formally accepted for publication once it meets all outstanding technical requirements.

Kind regards,

César Félix Cayo-Rojas, Ph.D.

Academic Editor

PLOS ONE

Additional Editor Comments (optional):

Thank you very much for responding to all comments. 

Reviewers' comments:

Reviewer's Responses to Questions

**Comments to the Author**

1. If the authors have adequately addressed your comments raised in a previous round of review and you feel that this manuscript is now acceptable for publication, you may indicate that here to bypass the “Comments to the Author” section, enter your conflict of interest statement in the “Confidential to Editor” section, and submit your "Accept" recommendation.

Reviewer #2: All comments have been addressed

2. Is the manuscript technically sound, and do the data support the conclusions?

Reviewer #2: Yes

3. Has the statistical analysis been performed appropriately and rigorously? 

Reviewer #2: Yes

4. Have the authors made all data underlying the findings in their manuscript fully available?

Reviewer #2: Yes

5. Is the manuscript presented in an intelligible fashion and written in standard English?

Reviewer #2: Yes

6. Review Comments to the Author

Reviewer #2: You have improved your work considerably, just one final recommendation, check in detail the use of abbreviations in the text and tables.

7. PLOS authors have the option to publish the peer review history of their article (what does this mean?). If published, this will include your full peer review and any attached files.

Reviewer #2: No

---

## [Editor Report · Acceptance letter]

2 Jun 2023

PONE-D-22-27517R3 

Influence of an oral health promotion program on the evolution of dental status in New Caledonia: a focus on health inequities. 

Dear Dr. Tubert-Jeannin:

I'm pleased to inform you that your manuscript has been deemed suitable for publication in PLOS ONE. Congratulations! Your manuscript is now with our production department. 

Kind regards, 

on behalf of

Dr. César Félix Cayo-Rojas 

Academic Editor

PLOS ONE